# RL-based Stateful Neural Adaptive Sampling and Denoising for Real-Time Path Tracing

**Antoine Scardigli, Lukas Cavigelli, Lorenz K. Müller**
Computing Systems Lab, Huawei Zurich Research Center, Switzerland
`scardigliantoine@gmail.com`, {`lukas.cavigelli, lorenz.mueller`}`@huawei.com`

## Abstract

Monte-Carlo path tracing is a powerful technique for realistic image synthesis but suffers from high levels of noise at low sample counts, limiting its use in real-time applications. To address this, we propose a framework with end-to-end training of a sampling importance network, a latent space encoder network, and a denoiser network. Our approach uses reinforcement learning to optimize the sampling importance network, thus avoiding explicit numerically approximated gradients. Our method does not aggregate the sampled values per pixel by averaging but keeps all sampled values which are then fed into the latent space encoder. The encoder replaces handcrafted spatiotemporal heuristics by learned representations in a latent space. Finally, a neural denoiser is trained to refine the output image. Our approach increases visual quality on several challenging datasets and reduces rendering times for equal quality by a factor of 1.6x compared to the previous state-of-the-art, making it a promising solution for real-time applications.

## 1 Introduction

Monte-Carlo path tracing relies on repeatedly sending discrete rays that bounce in random directions to approximate the real-life (continuous) light scattering. The number of rays sent per pixel is called the Sample Per Pixel (spp) count. The Monte-Carlo method guarantees that the estimated pixel value is unbiased and that the standard error decreases at a rate proportional to $\frac{1}{\sqrt{n}}$ where $n$ is the spp count [1, 2, 3, 4].

**Denoising**  Offline rendering can afford high spp counts. Nevertheless, it is very common to use spatial denoising as postprocessing to reduce the computational budget by several orders of magnitude for equivalent quality outputs [5]. First, denoising approaches were heuristic-based [6], but learned methods recently imposed themselves [7]. In particular, encoder-decoder neural networks like UNETs [8] are the current state of the art in terms of denoising [9]. Training the denoiser specifically for low sample counts (4 spp) allows still reaching high-quality results [10].

**Adaptive Sampling**  Following the intuition that low spp counts are likely sufficient for some parts of the frames like large uniform areas, and that some areas with bigger variance like edges or highly reflective surfaces might require a higher sample count, heuristic-based methods for sampling a non-uniform sample count within a frame have first been proposed. Some approaches use specific metrics like variance [11], contrast [12], frequency content [13], or mean absolute deviation [14] to assign more samples in challenging areas. Further work noticed that the denoiser should use information from the sampling heatmap or vice versa. They perform joint adaptive sampling and denoising by estimating the most problematic regions for the denoiser using Gaussians [15], linear regression [16], or polynomials [17]. As these metrics have to be estimated to sufficient precision, a high spp count (4 or more) is required, making them unsuitable for real-time applications: It is

37th Conference on Neural Information Processing Systems (NeurIPS 2023).

considered that real-time ray tracing corresponds to a rendering process with a latency smaller than 30 ms, assuming that roughly 30 frames are rendered per second. Such a computational budget generally corresponds to an overall spp count between 0 and 4 for a million pixels using a single recent+ GPU, depending on the scene to be rendered [18, 19].

**Gradient-based Adaptive Sampling** *Deep Adaptive Sampling for Low Count Rendering* [20] is the current state-of-the-art in adaptive sampling and the first to use a learned approach: The authors propose to first sample uniformly 1 spp and then use a UNET to generate the sampling heatmap for the remaining budget (3 spp). Finally, another UNET denoises the averaged sampled pixel values. In order to train the networks end-to-end, the gradient of the rendered image needs to be computed with respect to the sampling map for every pixel, which is not analytically possible. The authors estimate the gradient numerically instead:

$$\frac{\partial I_s}{\partial s} = \frac{I_\infty - I_s}{s} \tag{1}$$

with $I_\infty$ the ground truth pixel value, $I_s$ the average of the sampled pixel and s the sample count.

We identify 3 issues with this derivation: First, the numerical approximation converges to the real gradient in the limit, but no guarantees are offered for lower sample counts. Second, by design, the formula uses $I_s$ (the average of the sampled pixel) which enforces that sampled pixel values are aggregated by averaging them in the rest of the framework, hence surrendering information like higher-order statistics. Finally, the method does not allow sampling 0 spp for any pixel as this would make the gradient unstable.

*Deep Adaptive Sampling and Reconstruction using Analytic Distributions* [21] approximates the effect of the path-sampler using an analytical (gamma) distribution which allows backpropagating the gradient to the sampling importance network. This method is not suitable for real-time applications, but is very efficient training-wise because it only uses the ground truth estimate's mean and variance and does not use additional sampled frames.

*Adaptive Incident Radiance Field Sampling and Reconstruction Using Deep Reinforcement Learning* [22] leverages reinforcement learning to adaptively sample in the first bounce incidence radiance field by iteratively partitioning a tile into several tiles, or doubling the sample count of a tile. The first bounce incidence radiance field is then reconstructed and integrated with the BRDF for rendering. This method is unfortunately too slow for real-time applications.

**Spatiotemporal Reuse** Assuming that the goal is to render an animation and not a single frame, reusing information from previously rendered frames or from samples collected during the rendering of the previous frames can increase the quality of the result as it can increase the implicit sample count. Motion vectors: pixel-coordinate mappings of backward-warping, are used to warp the saved information from the previous frame so that it can be reused for the next frame.

Some related works only save the averaged pixel values of the previous frame [23], some save the averaged pixel values as well as some higher order statistics like the variance [24], and some store a subset of the sampled pixel values (non-averaged) into a *spatiotemporal reservoir* [25, 26] (a 3D grid that is used to store a list of path traced color samples for every pixel of the 2D image) to directly store an estimation of the true distribution. For example, ReSTIR [25] outperforms the state of the art in scenes with thousands of lights, and ReSTIR GI [27] in scenes where lights are seldomly visible through shadow rays. Those two methods interact with the path-sampler instead of considering it as a black box like other compared work.

**Neural Temporal Adaptive Sampling** [23] is the current state-of-the-art in real-time Monte-Carlo rendering using adaptive sampling. The authors reuse the gradient approximation of [20] to train the sampling importance network, but add a temporal feedback that is the previously obtained denoised averaged sampled pixel values. This temporal feedback is the main input of the sampling importance network, and one of the main inputs of the denoiser.

**Hybrid Latent Space** Using samples rather than pixel averages increases the computational cost but can improve denoising [28, 29, 30]. Some hybrid approaches try to get both benefits: [31, 32] are learned denoising (with uniform sampling) methods that propose a hybrid approach between keeping sampled pixel values and directly averaging them: Every individual sampled pixel value and its additional features go through a fully connected network, and only then are all latent frames an

aggregated average. This allows extracting some additional statistics than the average. Nevertheless, the numerical approximation of the gradient (Equation 1) requires to use the averaged pixels for backpropagating the gradient through the path-sampler, which prevents current adaptive sampling methods to use information other than the averaged sampled values.

We notice two opportunities along which the previous state-of-the-art in adaptive denoising [23] can be improved:

1. The spatiotemporal reuse only stores the previous denoised averaged frame and hence does not extract any other higher-order statistics than the average from the distribution of path-traced samples. This is problematic because information like variance or confidence is not used. For example, the sampling importance network has to guess the areas with high variance by learning to detect edges or highly reflective surfaces whereas the variance information could have been stored in the previous frame. One of our motivations is to increase the spatiotemporal-information reuse.

2. We notice that the numerical approximation of the gradient (Equation 1) that is derived in [20] has some shortcomings: such as being unstable for 0 spp counts, and being a rough numerical approximation that only converges to the true gradient in the limit. Our second motivation is to use Reinforcement Learning (RL) for adaptive sampling because it can learn an implicit gradient that works better than the approximated numerical one given our quantized problem.

## 2 Method

### 2.1 Spatiotemporal Latent Space

We mentioned that spatiotemporal reuse could consist of storing the average, some higher-order statistics, or directly the sampled pixel values to get the most insight of the sampled distribution. An example of this most efficient reusing is ReSTIR [25, 27] which updates a list of sampled colours for every pixel through probabilistic heuristics. To minimize the spatiotemporal loss of information, we use a spatiotemporal reservoir too (that we will call spatiotemporal latent space). However, instead of updating the spatiotemporal latent space using resampled importance sampling [33], we train a CNN network that learns to update the latent space in an optimal way.

The output of the latent state encoder network - the new spatiotemporal latent space - is the sole input of the denoiser and the main input of the sampling importance network. In comparison, previous work [23] gives the output of the denoiser to the sampling importance network. This is a waste of information because all the inputs of the denoiser are compressed into only 3 channels. A simple way to see the problem of this approach is that the denoiser network and the sampling importance network have no information in their inputs to get an insight about confidence: They do not have any input that stores the variance or the sample count. Finally, the fact that the latent space is the sole input of the denoiser leads to our method being very temporally stable without needing an explicit temporal loss as in [23].

### 2.2 Reinforcement Learning-based Adaptive Sampling

We described in the introduction section that the current state of the art for real-time adaptive sampling is based on deep learning using a numerical gradient approximation (Equation 1) for the sampling pass. We identified some problems with this numerical gradient approximation that could be mitigated using RL-based sampling recommendations:

1. The gradients are not analytically available: DASR [20] and NTAS [23] use an approximation of the real (inaccessible gradient) with the only guarantee that it converges to the true one in the limit, which is the opposite case than for real-time applications that use low spp counts. Instead of this explicit approximated gradient, it is possible that the RL algorithm will learn a better implicit gradient, allowing higher-quality learning and outputs.

2. Another limitation of the approximated gradient is that it does not allow to sample 0 samples per pixel because the numerical approximation is not stable in this case. For this reason, prior work [23, 20] cannot scale down to 0 spp. With the average spp budget for real-time

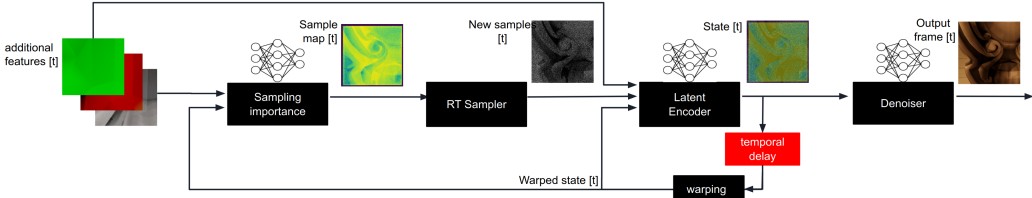

Figure 1: An overview of our Approach. The red temporal delay indicates that the warped state is delayed for the next time iteration. Warping takes as additional omitted input the motion vectors.

path tracing being in the range $[0; 4]$, this first means that those methods do not work when the spp budget is smaller than 1, and also that their sampling recommendations are very constrained in the remaining case. Using reinforcement learning allows us to not use this explicit gradient and hence avoid this issue.

3. Furthermore, the numerical approximation uses the averaged sampled pixel values for backpropagating the gradient through the path-sampler, which constrains the denoiser into using the averaged sampled pixel values instead of more complete information like higher order statistics or directly the sampled pixel values. Using RL removes this constraint.

## 2.3 Algorithms

Our approach uses 3 models:

**The Sampling Importance Network** is a UNET [8] that takes as input the additional features and the warped latent space, and outputs a sample map. The additional features have 7 channels (3 for the normals, 3 for the albedo, and 1 for the depth), and the latent space has 32 channels. Therefore, the sampling importance network takes 39 channels as input and one channel as output. The choice of the UNET architecture has two main motivations: First the upsampling will allow the output frame to contain sparse recommendations, and second, it makes sense that the recommendation values need to exploit information at different scales. The architecture is inspired by [8] because of computational efficiency. The actual recommendations $\vec{y}$ given the network recommendations $\vec{x}$ are as follows: $\vec{y}_{i,j} = \text{round}\left(\text{spp\_budget} \cdot \frac{\vec{x}_{i,j} - \min(x)}{\sum_{a,b}(\vec{x}_{a,b} - \min(x))}\right)$. We found that the final recommendations would become unstable if the network recommendations $x$ were not bounded because some of the values of $x$ would be dominantly larger than others. We thus bound the network recommendations by adding a Tanh activation layer [34] as final layer. The number of output channels of the first convolution (4) is much smaller than the input number of channels (39), following the rationale that most of the encoded information in the latent space is mostly useful for the denoiser network but not for the sampling importance network.

**The Latent Encoder Network** is a CNN [35] that takes the warped latent space and the new samples as input, and outputs a new latent space. New samples can contain up to 8 values per pixel. For this reason, the new-samples input contains 24 channels, where non-attributed pixel values are assigned the value (-1) with the rationale that the path-sampler only outputs non-negative values. The 8 frames are such that if a pixel is assigned a sample count of $n$, the first $n$ frames will contain sampled pixel value, and the last $8 - n$ frames will contain the anomalous value "-1" for this pixel coordinate. This architecture is not permutation invariant, which allows it to extract information from all sampled values of the same pixel and from spatially neighboring pixels unlike previous work in hybrid latent space [23, 31, 32]. We added a Tanh layer at the end of our network to bound the state values. The number of state channels (32 in our case) is chosen considering the performance over time trade-off.

**The Denoiser** is a UNET that takes the new latent space as input and outputs a denoised image. Using a UNET for ray tracing denoising is a recommended design [10], and we reuse the architecture from [9] as it is one of the most widely used solutions in terms of fast ray tracing denoising. We do not use kernel prediction [7] because [23] found it was not significantly useful.

In the Appendix, we present a network variational study that motivates the size every network should have to maximize PSNR at a given latency, and we include diagrams of the network architectures.

## 2.4 Loss, RL Algorithm, and Reward

We train the latent encoder state network and the denoiser network using gradient backpropagation, and the sampling importance network using RL-based policy optimization leveraging APPO [36]. Given the ground truth image, we backpropagate the gradients for the denoiser and the latent space using the mixed $\ell_1$-MS-SSIM loss [37] with a ratio of 0.16-to-0.84. The reward for the sampling importance network is set to be ten to the power of one minus the loss. Our RL framework can be represented by $(S, A, P, R, \gamma)$, with $S$ the observation space, $A$ is the action space, $P$ is the transition probability function from action and old observation to new observation, $R$ is the reward function, and $\gamma$ is the discount factor. Given that $f$ is the denoiser, $g$ is the latent state encoder, $h$ is the ray sampler, and $\omega$ is the time warper: $S$ is in the space $[-1; 1]^{720 \cdot 720 \cdot 39}$. $A$ is in the space $[-1; 1]^{720 \cdot 720}$. $P(s'|s, a)$ follows the distribution of $(\omega \circ g)(h(a), s)$. $R(s, a, s')$ is $10^{1-loss((f \circ g)(h(a),s),gd)}$ where $gd$ is the corresponding ground truth image, and $loss$ is the mixed $\ell_1$-MS-SSIM loss. $\gamma = 0.99$.

## 2.5 Training

All networks are trained in a closed loop on 100 epochs with a batch size of 4. We train all networks end-to-end, such that the sampling importance network can insert more samples where the denoiser does a poor job, and vice versa. The learning rate schedule follows the same pattern as the learning rate used to train the OIDN Denoiser [9]: the learning rate has a maximum value of 0.1 and a minimum value of $10^{-8}$, with a warmup phase of 15%, and then an exponential decay phase. We use Adam optimizer [38], Pytorch and Ray-RLlib [39]. We transform the images by using vertical and/or horizontal flips, by randomly cropping and rescaling the images. We randomly permute the order of the input images to teach the state encoder to be permutation invariant.

**Baselines** We present in Table 1 the description and inference time of every approach. Unlike our work, some of the presented baselines did not release their implementation [23, 24, 25, 31], we therefore reimplemented them as described by the authors or used unofficial implementations, and trained all algorithms using the same amount of data and epochs.

Table 1: Description, visual quality evaluation at 4.0 spp budget, and inference time in ms.

| Algorithm name | Description | 4.0 spp PSNR (dB) | Inference time (ms) |
| --- | --- | --- | --- |
| ours | Our method | 28.7 | 22.5 |
| ours-A1 | Ours, no RL, but gradient approx. | 28.4 | 22.5 |
| ours-A2 | Ours with uniform sampling | 27.9 | 20.0 |
| ours-B1 | Ours, no latent space encoder | 28.0 | 19.0 |
| ours-B2 | Ours-B1, no temporal feedback | 27.4 | 18.8 |
| ours-C | Ours with averaged ray samples | 28.2 | 22.1 |
| ours-small | Ours scaled-down version | 27.3 | 14.4 |
| IMCD | [31] | 27.9 | 46.4 |
| NTAS | [23] | 27.7 | 17.4 |
| DASR | [20] | 27.2 | 21.9 |
| ReSTIR+OIDN | ReSTIR denoised by OIDN [25, 9] | 28.0 | 18.3 |
| SVGF | [24] | 22.8 | 3.9 |
| OIDN | [9] | 26.2 | 16.3 |
| ReSTIR | [25] | 22.1 | 2.0 |
| MC | Monte-Carlo path tracing average | 17.0 | 0.0 |

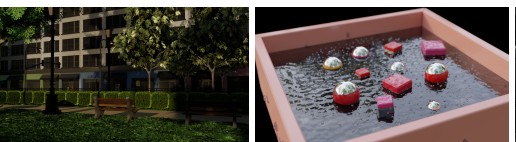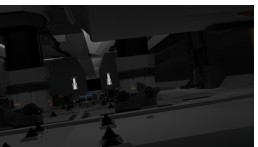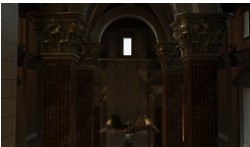

Figure 2: A ground truth image from the Emerald scene, the Ripple Dreams scene, the ZeroDay scene, and the Suntemple scene (going left to right)

# 3 Results

## 3.1 Experimental Setup

As in related work [23, 20, 24], we precompute a dataset such that live communication with the path-sampler is not necessary. We use the path-sampler **Cycles** using the GPU's RTX acceleration. We use three 3D scenes *Emerald Square* [40], *SunTemple* [41], and *Zero-Day* [42] released as part of the Open Research Content Archive (ORCA) under CC BY 4.0 license. We also use the scene *Ripple Dreams* released by James Redmond. The scenes we used from the ORCA project are the biggest available subset of the scenes used in the most related work [23]. Unfortunately, the ORCA scenes have been further modified and not shared by the authors of [23], which makes an exact usage of their scenes impossible. We release our dataset scenes and code implementation[1] to facilitate the usage and comparison of our method.

For every 3D scene, we extract a sequence of frames that forms a continuous video animation. For the Emerald scene we extract 1400 frames, for the ZeroDay scene we extract 400 frames, for the SunTemple scene we extract 1200 frames, and for the Ripple Dream scene we extract 500 frames. We further split those animations into independent clips of 20 frames. For every frame, we compute the ground truth image which is an unbiased Monte-Carlo estimate using 1000 spp, the additional data which includes normals, albedo, depth, motion vectors (a pixel-to-pixel mapping from one frame to the next one), and the inputs which are 8 path-traced images with 1 spp each. Previous work [23, 20] instead stored input images with spp count $2^i$ for $i = 0$ to 5, such that they could compute the average pixel values in the range $[0; \sum_{i=1}^{5} 2^i] = [0; 63]$ using only 5 input frames, whereas our method gives access to between 0 and 8 pixel values instead of only the average. The limit of 8 samples has been chosen as we have seen that less than 0.6% of the pixels get higher spp count recommendations in 4 spp average count scenarios. For scenarios with higher spp rendering counts than 4 (which would not be realistic in practice for real-time usage), increasing this number could be required at the cost of increasing GPU memory usage.

We perform our measurements on a Nvidia GeForce RTX 2080 Ti GPU. The sampling times for 1 spp frames with a resolution of $720 \times 720$ are shown in Table 2.

## 3.2 Quantitative Results

In the result section, we will present the cross-validated results for each scene: For every test scene, we train on all the other scenes, and output the average PSNR over the whole test scene. We use PSNR as metric because this is the most widely used metric used in the relevant literature [23, 27, 20]. Instead of presenting the PSNR over latency time per frame for a single spp count as in the related literature, we include several different spp counts to get an insight into the quality over time tradeoff for every method. in this section we present the averaged results, but the cross-validated results per dataset are available in the Appendix. We measure a standard deviation of 0.06 dB for our method based on 5 trainings with different seeds, which makes barplots or confidence intervals unnecessary.

## 3.3 Qualitative Results

We present in Figure 4 a qualitative example of the visual quality of our method's rendering compared to the previous state of the art. Please look at the additional material to observe more qualitative results including videos.

---

[1]Source code available at `https://github.com/AJSVB/RL_PATH_TRACING`.

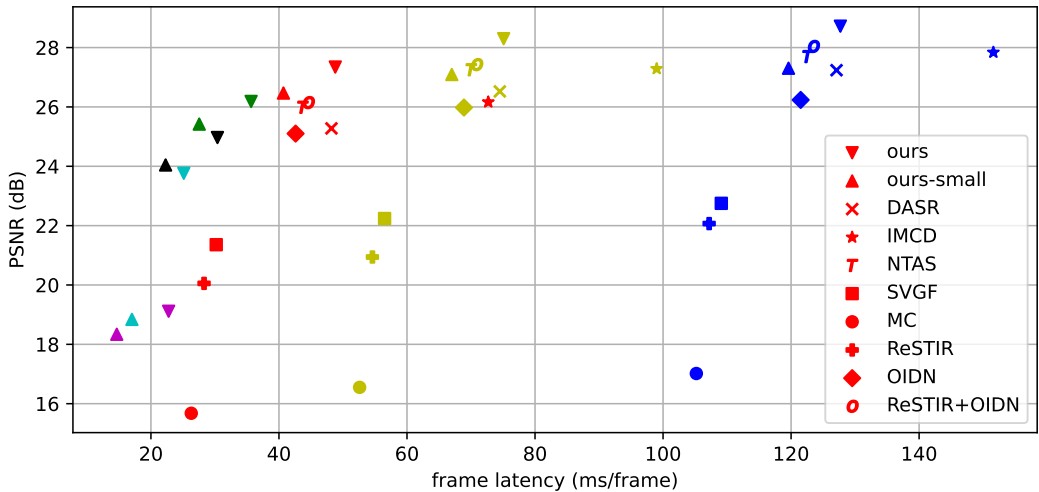

Figure 3: PSNR quality as a function of frame latency for the overall cross-validated averaged results among all datasets. Magenta color corresponds to a 0.01 spp count, cyan color corresponds to a 0.1 spp count, black color corresponds to 0.3 spp count, green color corresponds to a 0.5 spp count, red color corresponds to a 1.0 spp count, yellow color corresponds to a 2.0 spp count and blue color corresponds to a 4.0 spp count. Upper-left is better.

Table 2: Sampling time for 1 spp given our datasets and hardware.

| Dataset | 1 spp sampling time (ms) |
|---|---|
| Suntemple | 16.6 |
| Ripple Dream | 17.6 |
| Emerald | 34.5 |
| ZeroDay | 36.6 |
| Average | 26.3 |

## 4    Discussion

**Impact of the Sample Count**    In Figure 3, we evaluate the tradeoff between frame latency and PSNR and compare it to previous methods for various sample counts. Notice that only adaptive methods can use non-integer spp budgets because there is no procedure to choose how to sample non uniformly for non-adaptive methods. Previous adaptive methods approach uniform sampling as their spp budget approaches 1.0 because they cannot sample less than 1 spp for any pixel. Indeed, the gradient approximation (Equation 1) diverges in case the spp recommendation is 0. This imposes a hard lower limit on the frame latency. Contrapositively, only our method can use a spp budget smaller than 1.0 because we train the sampling importance network with reinforcement learning to overcome the diverging gradient limitation. We observe that the visual quality collapses for very low spp counts (0.01 spp), greatly increases from 0.01 to 2 spp, and then increases slower until 4 spp count.

We present a variant of our method called "ours-small" that corresponds to our method with smaller sampling importance, latent encoder, and denoiser networks (see the network variational study in the Appendix). This variant has a smaller inference time which allows reaching even lower frame latencies. We observe that this variant is Pareto-optimal until 38 ms/frame. We also notice that this variant really starts improving after reaching a 0.1 spp count, whereas our main method already improves between 0.01 spp and 0.1 spp counts. This indicates that our variant with small networks is not able to fully leverage the potential of 0.1 spp count.

**Comparison of our Method with other Related Work**    We observe that our method is Pareto-optimal compared to previous methods for any spp budget. Not only does our method outperform any other baseline for equal spp counts, but also for higher spp counts: our method with 2 spp budget outperforms every baseline with 4 spp budget. Given that some of the current trends in rendering aim

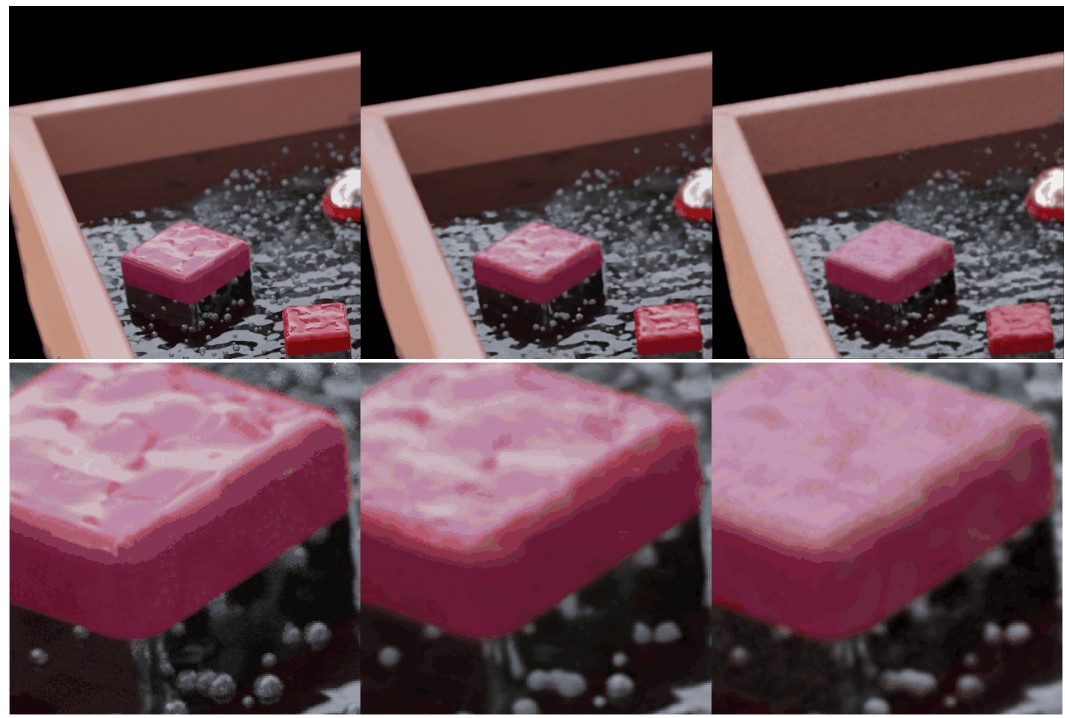

Figure 4: Ground truth using 1000 spp on the left, ours using 4.0 spp on the middle, previous adaptive sampling state-of-the-art (Neural Temporal Adaptive Sampling [23]) with 4.0 spp on the right. Image rendered on the Ripple Dream dataset. 720x720 frame on top, and 200x200 zoom to a particular area on the bottom. The PSNR for our rendered image is 29.1 dB and is 27.7 dB for the previous state-of-the-art rendered image.

at minimizing the latency time (no adaptive sampling, supersampling,...), the fact that our method has a relatively large latency but manages to outperform quality-wise very fast baselines for an equal latency is an interesting result [27, 24, 25]. Given an equal budget of 4 spp, our method renders outputs with a 0.6 dB higher PSNR compared to the strongest baseline for a negligible latency increase (3 ms latency increase which corresponds to 5.5%). Our approach with 2 spp reaches a higher visual quality (PSNR improvement of 0.2 dB) for a 1.6x latency reduction compared to the strongest baseline at 4 spp.

**Ablation Study of our Method**    We observe on Table 1 that our method is among the slowest in terms of ms/frames, but generally outperforms all other baselines. Our method and three of our variants (ours-A1, ours-B1, ours-C) always outperform all previous works except ReSTIR+OIDN [25, 9] for an equal spp count. On average, our method outperforms the variant with the gradient approximation (ours-A1) by 0.3 dB, and the variant with uniform sampling (ours-A2) by 0.8 dB. This demonstrates both the utility of adaptive sampling and the superiority of training the sampling importance network with RL instead of the gradient approximation. We include a study of the sampling heatmap and error distribution for several scenes in the supplementary material to improve the understanding of the effect of RL-based sampling vs gradient-based sampling.

Comparing our method to ours-B1, we see that using the latent space encoder and having as temporal feedback the warped latent space instead of the sampled pixel values (ours-B1) allows increasing the PSNR by 0.7 dB, and increases the PSNR by 1.3 dB in the case with no temporal feedback (Ours-B2). Finally using the sampled pixel values instead of the averaged values (ours-C) increases the PSNR by 0.5 dB.

**Comparison with Adaptive Sampling SOTA**    Our method using RL for the adaptive sampling outperforms our method using the gradient approximation (Ours-A1) for the adaptive sampling. Furthermore, our method with the gradient approximation outperforms NTAS [23] by a large margin

(0.7 dB) showing the importance of using the individual sampled pixel values instead of the averaged values, and of using a latent space encoder.

**Comparison with Hybrid Latent State SOTA**    Our method with uniform sampling (Ours-A2) has as high visual quality as IMCD [31]. This is not surprising because both approaches use temporal feedback and use sampled pixel values instead of only the average. The main difference in the design is our state encoder that creates a meaningful state, whereas IMCD only uses simple heuristics like an exponential average of the previous latent spaces. This difference allows our variant to reach equal visual quality despite the 2.1x smaller inference time.

**Comparison with Probabilistic Latent Space SOTA**    ReSTIR+OIDN [25, 9] uses probabilistic heuristics and the non-averaged sampled pixel values to recommend from which light to sample, and to choose how to update the spatiotemporal latent space. Our method conceptually has the same objectives. The fact that our method outperforms ReSTIR+OIDN by 0.5 dB shows that using networks that learn how to do the sampling and how to update the spatiotemporal latent state instead of using theoretical heuristics allows reaching higher results.

**Visual Quality Comparison**    Looking at Figure 4, we observe that our method generally looks more similar to the ground-truth image. For example, large uniform surfaces present less noise and advanced details such as edges, small bubbles, reflection, and refraction effects look more realistic with our method. In the supplementary material, we compare rendered animations and observe that our method manages to maintain temporally more coherent results as well.

**Limitations**    The current method does not consider the application of after-effects such as motion blur, which could allow the reduction of samples collected on fast-moving objects, while in the current setting we would implicitly focus the ray tracing samples on such an object to minimize the error. Further, this method is applicable to entertainment applications and can potentially generate artifacts not present in the real scene, which could be an obstacle in application scenarios such as VR/AR medical devices. Additionally, it requires a system capable of performing both path tracing and DNN inference. While current graphics cards provide this capability and we assess the frame rendering time considering both components, the underlying workloads are significantly different: DNN inference is generally a very structured and high arithmetic intensity workload whereas path tracing is branching-intensive and performs random look-ups into memory. As these are fundamentally different, we see dedicated ray tracing units on modern GPUs. In future systems, it is conceivable that dedicated devices are used for each step, which would enforce a fixed capability for each type of compute and limit a free trade-off between the two as we make use in this work. Additionally, applying a DNN adds a memory overhead, although it can remain minimal compared to other components such as textures that commonly fill most the GPU's memory. Specifically, the model requires 110 MB more memory to store the input and latent space data, <50 MB to store the model, and a few MB of working memory for intermediate feature maps that can be processed in tiles.

**Memory Overhead**    We store the following data in memory. 1) the model weights (<50MB), 2) the sampled pixel values (24 channels; RGB values for up to 8 non-averaged samples), 3) 7 additional input channels (3 for surface normals, 3 for albedo, 1 for depth), 4) 32 channels for the state (warped latent space), and 5) a small working memory for intermediate feature data during inference. The components 2-4 add up to 53 more channels than previous works (which use 3 channels for input channels and 7 for additional data); hence 212 Byte/pixel; 110MB in total for 720x720 pixel data; and the required working memory is minimal as the inference can be done on tiles. With a total memory footprint in the order of 200 MB, the overhead is insignificant (<2%) compared to >12GB of textures loaded for current games on high-end GPUs where such capacity is available.

## 5    Conclusion

We explored three different ideas to improve sampling and denoising for low sampling count and showed through a leave-one-contribution-out that they are all important: The first one is to adaptively sample leveraging RL to make better sampling decisions; using the previously recommended gradient approximation leads to a 0.3 dB PSNR decrease. The second one is to let a deep learning latent state

encoder decide which pixel information to keep given the previous state to optimize the denoising and further rendering; removing this network leads to a 0.7 dB PSNR decrease. The third one is to not aggregate the pixel values by averaging, but to keep all information for the state encoder network; averaging the sampled pixel values leads to a 0.5 dB PSNR decrease. We discovered that combining those ideas allows outperforming different state-of-the-arts by at least 0.7 dB for an equal spp budget, or by a 1.6x latency reduction while keeping a marginal (0.2 dB) visual quality improvement. Finally, we are the first adaptive sampling method that can accept a lower spp budget than 1 spp, allowing using adaptive sampling for unprecedentedly high frame rates. Finally, we release our implementation and datasets.

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

## A Individual Results Per Scene

The analysis in Section 3.3 averages the latency and PSNR across different scenes. This could potentially hide differences in the scene complexity where the cost of sampling a ray might vary significantly. We thus provide a more fine-grained per-scene evalautions in Figure 5.

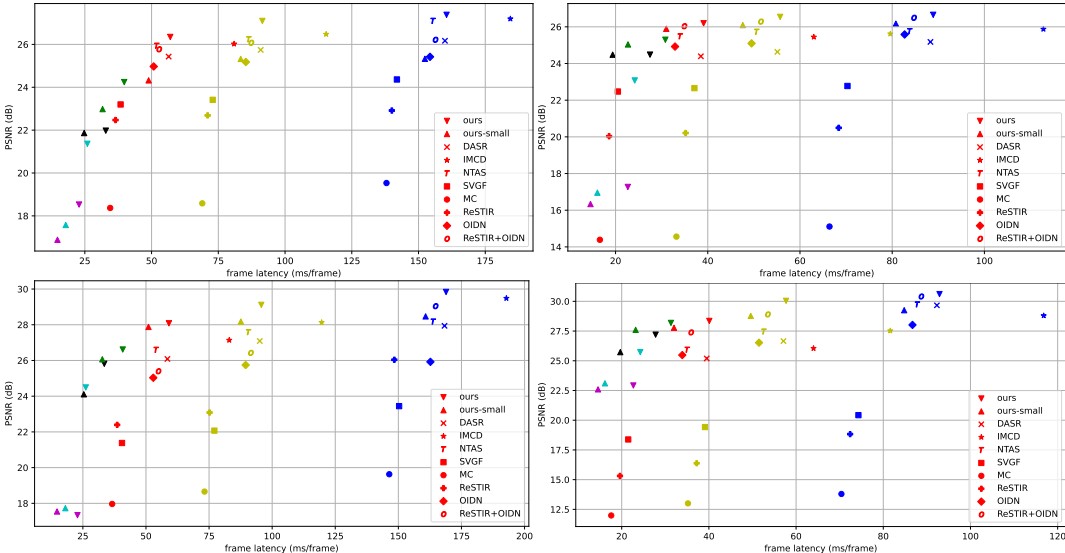

Figure 5: PSNR quality as a function of time per frame on the cross-validated scenes EmeraldPlace (top left), SunTemple (top right), ZeroDay (lower left) and Ripple Dream (lower right). Magenta color corresponds to a 0.01 spp count, cyan color corresponds to a 0.1 spp count, black color corresponds to 0.3 spp count, green color corresponds to a 0.5 spp count, red color corresponds to a 1.0 spp count, yellow color corresponds to a 2.0 spp count and blue color corresponds to a 4.0 spp count. Upper-left is better.

## B Architecture and Variational Network Study

The basic architecture of our networks is visualized in Figure 7. We justify the choice of size of the different different models used within our solution using a variational study: we fix a time budget of 100 ms per frame and select a single scene such that the sampling time for 1.0 spp is fixed. We then study how varying the size of each of our 3 networks affects the quality of the results, taking into account that changing the size of the network will change the total sampling budget. We evaluate our method on the EmeraldPlace dataset.

For the small variant of the denoiser we scale down the architecture shown in Figure 7 by reducing the number of channels by $2\times$ every convolution layer, and its large variant is identical except we scale up the number of channels by $2\times$. The small sampling importance network is a simple CNN with three $3 \times 3$ convolutional blocks using 1 latent channel, and the large sampling importance network is the original UNET from [8]. The small state encoder architecture corresponds to the architecture in Figure 7 except we replaced the $3 \times 3$ convolutions with $1 \times 1$ convolutions and the big state encoder architecture corresponds to the architecture in Figure 7 except that we added two additional $3 \times 3$ convolutions with a ReLU layer in between. Ours-small mentioned in the paper corresponds to using both the small sampling importance network, the small state encoder, and the small denoiser network.

**Sampling Importance Network**    This variational study shows that choosing a very fast sampling importance network is beneficial compared to using a slower architecture for our fixed time budget experiment. This result probably does not hold when the time budget increases because the inference time of the network will become negligible. Nevertheless, we focus on real-time applications which means that the time budget has to be smaller or equal to 30 ms. This architecture design contrasts with previous designs in adaptive sampling [23, 20], where the sampling importance network and the

Table 3: Inference time in ms and corresponding spp count for every network. Spp count is computed as follows: given x the total inference time, spp count = (100-x)/69.0, where 69.0 is the sampling time in ms for 1 spp on EmeraldPlace.

| Variation | Sampling importance | State encoder | Denoiser | Total | Spp count |
|---|---|---|---|---|---|
| Normal | 2.5 | 3.7 | 16.3 | 22.5 | 1.12 |
| Small Sampling importance | 2.0 | 3.7 | 16.3 | 22.0 | 1.13 |
| Large Sampling importance | 4.5 | 3.7 | 16.3 | 24.5 | 1.09 |
| Small state encoder | 2.5 | 2.2 | 16.3 | 21.0 | 1.14 |
| Large state encoder | 2.5 | 9.4 | 16.3 | 28.2 | 1.04 |
| Small denoiser | 2.5 | 3.7 | 10.2 | 16.4 | 1.21 |
| Large denoiser | 2.5 | 3.7 | 25.1 | 31.3 | 0.99 |

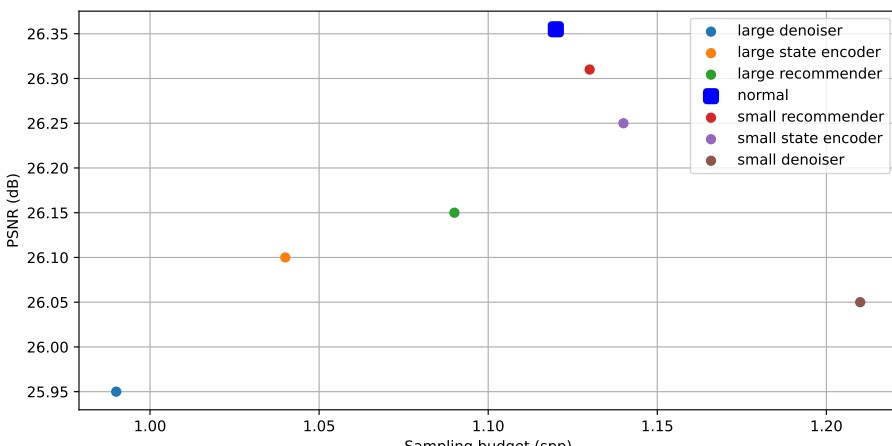

Figure 6: Comparison of PSNR when varying the size of every network for equal latency constraints. Higher is better.

denoiser network had almost the same architecture and inference time. This difference in design can be explained because we take the latent space as input which is pre-processed information, whereas previous approaches give row information as input, and the sampling importance network typically had to guess which were the high variance areas, for example by learning to segment edges, whereas our method can more easily extract more meaningful data like sample count, or variance from the latent space.

**State Encoder Network**    We also observe that the state encoder does not need to be a UNET, which implies that local information is sufficient to update the latent state.

**Denoiser Network**    Finally, the denoiser network takes the majority of the inference time of our approach. Our architecture is almost identical to the denoising architecture of [9]. We observe in the ablation study that trying to shrink this architecture strongly deteriorates the results.

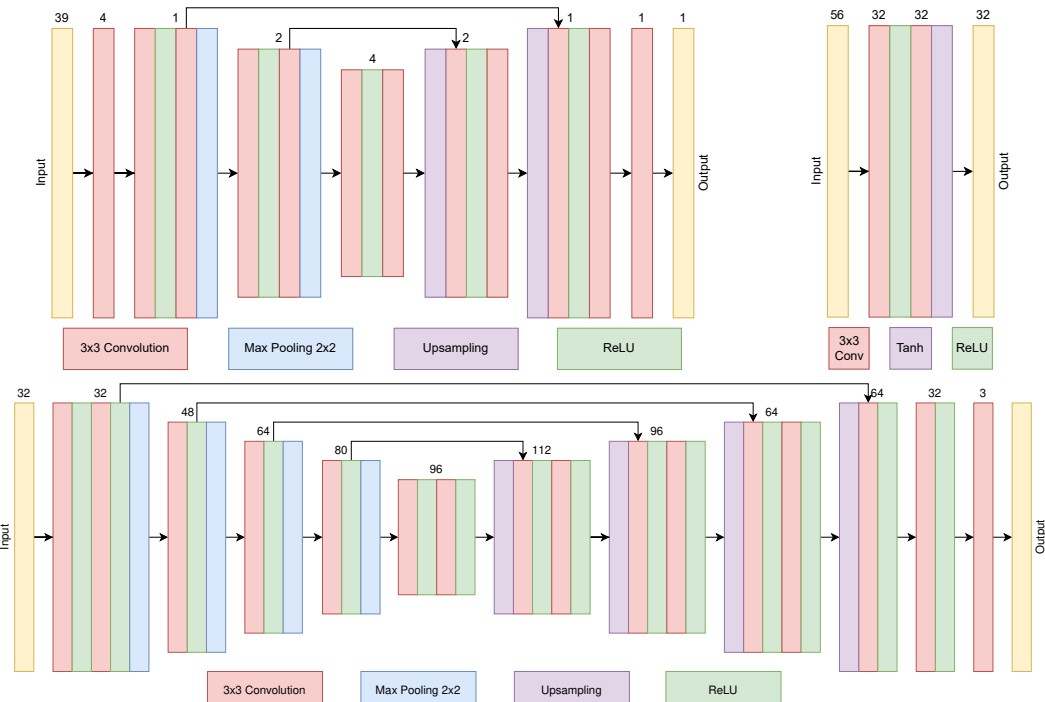

Figure 7: Architecture of the sampling importance network (top left), the latent state encoder network (top right), and the denoiser network (bottom). Max Pooling $2 \times 2$ refers to pooling with kernel size 2 and stride 2. Typically, given an input image with a resolution of $720 \times 720$, the resolution after the first max pooling is $360 \times 360$, after the second is $180 \times 180$, after the third is $90 \times 90$, after the fourth is $45 \times 45$, and the symmetrical opposite resolutions for the upsampling way. The number above the convolutional blocks represents the number of output channels for every layer in the block. Upper arrows represent residual connections. The concatenation layers are not shown but happen after every upsampling layer.

## C   Sampling Strategy Comparison

Identifying the differences between the sampling heatmaps when using reinforcement learning and when using the gradient approximation in non-trivial (Figure 10). We attribute this to the sampling importance network learning to adaptively sample by taking into account the impact of the denoiser, and not simply by sampling in high variance areas like edges. We found it more informative to compare the MSE per pixel between the RL adaptive sampling and gradient approximation adaptive sampling method (Figure 8,Figure 9). In the additional material, we include similar data for moving animations. The per-pixel difference in MSE between the methods remains challenging to interpret directly (Figure 8). To otherwise visualize the per-pixel MSE gains, we generated histograms (Figure 9) clearly showing that the RL-based method leads to generally better results as the mode and the mean of the distributions are always negative. In particular, we observe from the lower histogram that our RL-based method particularly avoids large errors.

**Sparse Sampling**   A common strategy that we observe in both our method using RL and using the gradient approximation, but that we did not observe in previous adaptive sampling methods is *sparse sampling*. Given a uniform area, sparse sampling consists of a pattern such that few pixels get high spp count recommendation, and surrounding pixels get low spp count recommendation. This can be an efficient sampling strategy assuming that locally close pixel values are not independent (cf. Figure 10).

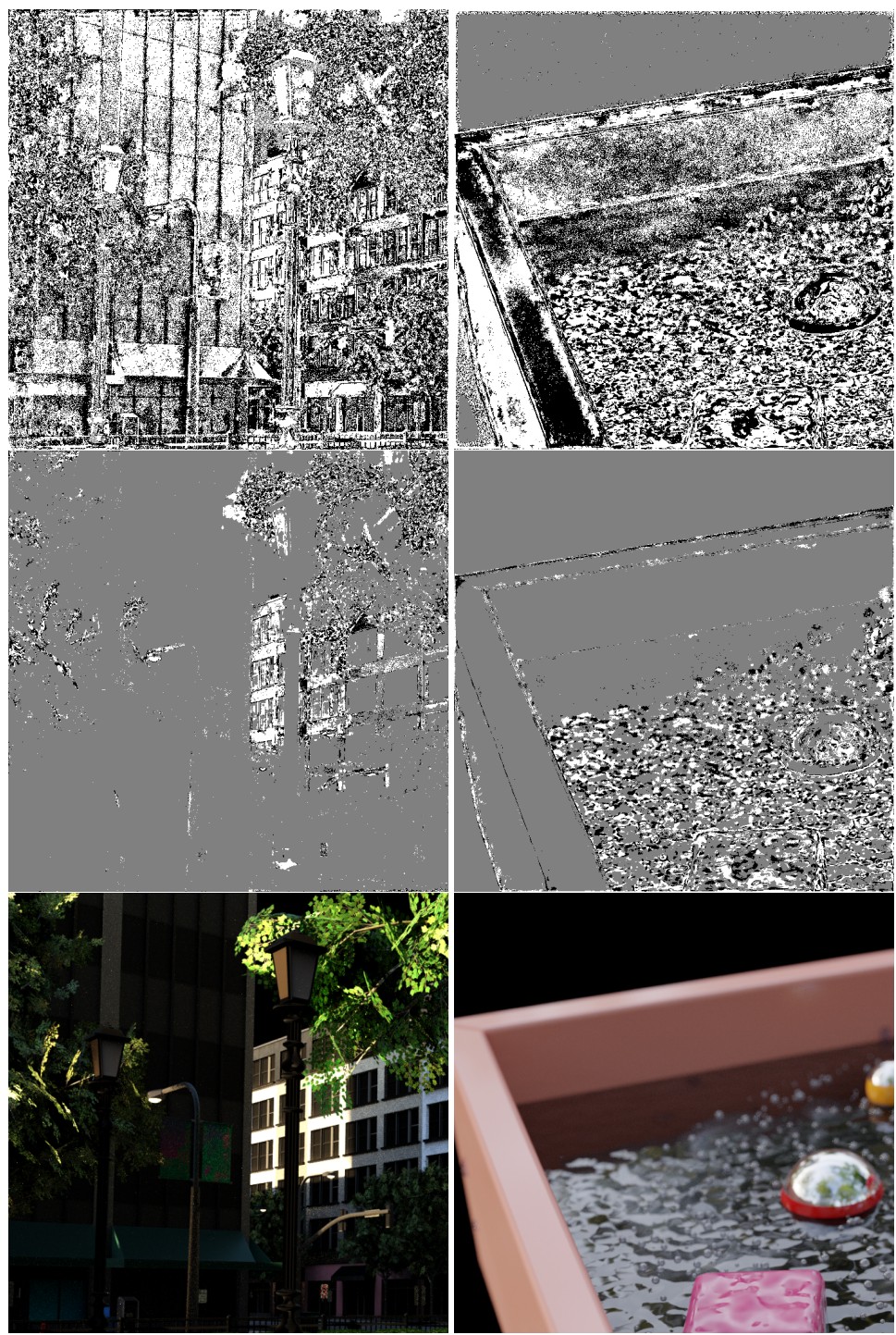

Figure 8: Comparison of the binary MSE difference between our method when trained with RL (top left), and the gradient approximation (top right). White colors mean that the MSE of the method with the gradient approximation is bigger for this pixel and inversely for black colors.

Comparison of the extreme binary MSE difference between our method when trained with RL (center left), and the gradient approximation (center right). White colors mean that the difference in MSE of the method with the gradient approximation is bigger than the threshold for this pixel, and inversely for black colors.

Ground truth images included for reference (bottom left and bottom right).

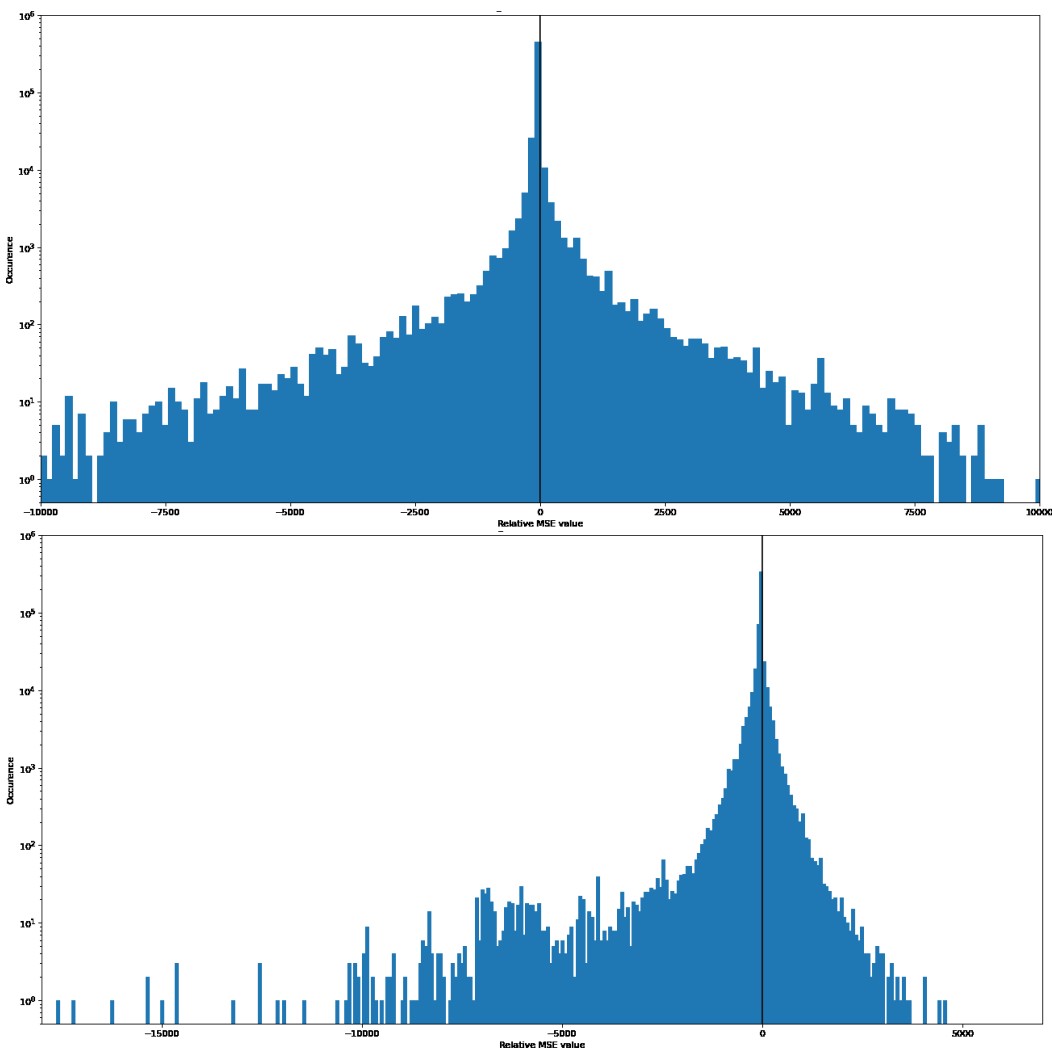

Figure 9: Comparison of the distribution of MSE differences between our method when trained with RL and with the gradient approximation. Negative values mean that the MSE of the method with the gradient approximation is bigger and inversely for positive values. The top image corresponds to the distribution for the left image in Figure 8, and the bottom image to the distribution for the right image in Figure 8.

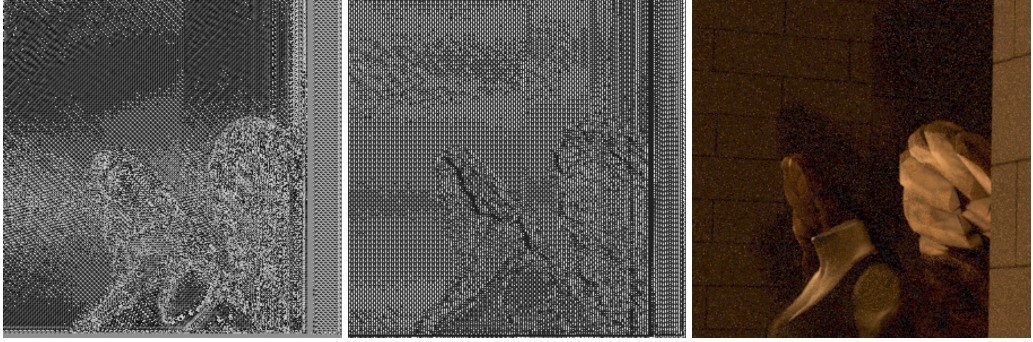

Figure 10: Comparison of the sampling recommendation from our sampling importance network when trained with RL (left), and the gradient approximation (center). Lighter colors mean higher sampling recommendations. Identical color scale. We observe sparse sampling and recurrent sampling patterns in both sampling heatmaps. Ground truth image included for reference (right).

## D    Supplementary Material

Together with the paper, we provide several short video sequences to verify that our solution does not introduce any flickering or other artifacts and generally enable a qualitative comparison of methods. *GD-ours-NTAS.gif* extends Figure 4 in the main manuscript. We include *FIG4a.gif*, *FIG4b.gif*, *FIG4c.gif*, *FIG4d.gif*, and *FIG4e.gif* that extend Figure 8 of this manuscript. We include *FIGupper.gif* and *FIGlower.gif* that extend Figure 9 from this manuscript. The GIFs have been compressed to max. 100MB.

