# OpenReview forum: "RL-based Stateful Neural Adaptive Sampling and Denoising for Real-Time Path Tracing"
_NeurIPS.cc/2023/Conference — NeurIPS 2023 poster_

### Official Review · Reviewer_YEyS · 2023-07-03

**Soundness:** 3 good
**Presentation:** 3 good
**Contribution:** 3 good
**Rating:** 5
**Confidence:** 3

**Summary:**

Ray tracing faces the difficulties of being applied practical real-time applications due to high levels of noise when sample counts are low. Sample counts are often limited to as low as 4 when considering real-time applications (~30ms latency). As such, the paper proposes an end-to-end training of a RL-based sampling importance network, a latent space encoder network, and a denoiser network. As a result, the proposed framework achieves strong performance on several benchmarks when considering performance and latency trade off.

**Strengths:**

- The paper is well-written and easy to follow
- The proposed framework demonstrates strong performance in terms of performance and latency trade off.

**Weaknesses:**

- What is the exact novelty in comparison to previous works? The proposed framework seems to be combination of RL-based adaptive sampling [22], the use of sampling importance network [23], spatiotemporal reservoir (spatiotemporal latent space) [25,27], and the use of denoisers [10], except for minute details, such as not feeding the output of a denoiser to the sampling importance network?

- Without detailed discussions on the major differences between the proposed method and previous works and/or combination of previous works, it is difficult to assess the novelty and contribution of the proposed method.

- How does the memory consumption compare between the proposed method and previous works?

- Is the inference time in Table referring to the latency of the whole framework, including the inference of networks?

**Questions:**

Please refer to the weakness section.

**Limitations:**

The paper does not include the limitation section. One limitation would be slightly higher inference time compared to previous works. Another possible limitation would be larger memory consumption, which cannot be said for certain until the authors provide comparisons.

---

> ### Author Rebuttal · Authors · 2023-08-08
>
>
> Dear Reviewer YEyS, thank you for your time and interesting remarks.
>
> > What is the exact novelty in comparison to previous works? The proposed framework seems to be combination of RL-based adaptive sampling [22], the use of sampling importance network [23], spatiotemporal reservoir (spatiotemporal latent space) [25,27], and the use of denoisers [10], except for minute details, such as not feeding the output of a denoiser to the sampling importance network? Without detailed discussions on the major differences between the proposed method and previous works and/or combination of previous works, it is difficult to assess the novelty and contribution of the proposed method.
>
> We have 2 major novelties compared to previous work (cf lines 108-144):
>
> 1) Learned latent space representation: The only previous works using a latent space are ReSTIR and derivative works. Their latent space consists of storing a buffer of lights for every pixel. They update the buffer by sampling from new visible lights or reusing spatiotemporally nearby pixels, leveraging simple statistical heuristics (importance sampling). This latent space is directly used only for denoising.
> Differently, our latent space consists of storing a learned encoding given all previously available information to guide future sampling and denoising. It is updated using a dedicated neural network given new information and the previous state.
>
> 2) RL-based learning of the sampling recommendation model:
> The only related work using RL is [22], which is not suitable for real-time and does not deal with adaptive path tracing but with the incompatible task of incidence radiance field.
> Previous work targeting sampling recommendation (DASR [20], NTAS [23]) used approximated gradients instead of RL. We avoided such an approximation and instead formulated an exact end-to-end optimization criterion using reinforcement learning. It also allowed us reaching unprecedented low sample counts and hence inference times as non-RL methods naturally cannot be scaled below 1spp as there is no longer a gradient when no samples are collected.
>
> To identify the impact of each component, we perform an ablation study (Lines 277ff. and Table 1) showing that each contribution is essential to achieve the final performance.
>
> > How does the memory consumption compare between the proposed method and previous works?
>
> The memory usage from neural networks weights is minimal (<50MB). We store more input and state information than previous work: besides the sampled pixel values (24 channels; RGB values for up to 8 non-averaged samples) and 7 additional input channels (3 for surface normals, 3 for albedo, 1 for depth), we store 32 channels for the state (warped latent space)---a total of 53 channels more than previous work (using 7 channels for additional data and 3 channels for pixel values) or 212 Byte/pixel which amounts to 110 MB at 720x720 pixels. As working memory during inference, we can operate with a minimal buffer size as the high-resolution images can be tiled. With a total memory footprint in the order of 200 MB, the overhead is insignificant (<2%) compared to >12GB of textures loaded for current games on high-end GPUs where such capacity is available.
>
> > Is the inference time in Table referring to the latency of the whole framework, including the inference of networks?
>
> In Table 1, we report the execution time of the method without the cost of collecting ray tracing samples, i.e., the neural networks and warping for our method or the best implementation we could find of related work such as ReSTIR with Monte-Carlo path tracing setting the baseline at 0 inference time. We report these as they are largely independent of the scene. In Figure 3, we compare the latencies of the entire systems including networks and collecting the pixel samples, evaluated on the test datasets.
>
> > The paper does not include the limitation section. One limitation would be slightly higher inference time compared to previous works. Another possible limitation would be larger memory consumption, which cannot be said for certain until the authors provide comparisons.
>
> The main purpose of our paper is to deliver a better PSNR/inference-time trade-off. Our main results (Figure 3) show the relationship between frame-latency (which is the inference time of the whole framework including the pixel sampling) and the quality for our method (including variants) and relevant related work. Our method is Pareto optimal (always on the upper left), which means it has a lower inference time for equal visual quality compared to all other methods. Our method has a marginally larger memory consumption.
> Based on the feedback of Reviewer WTeK, and if permitted, we will add a "limitation" section to the paper or at least to the appendix (see official comment).
>
>
> Thank you for dedicating time to evaluate our paper and provide thoughtful feedback. We appreciate your acknowledgment of its strengths in soundness, presentation, and contribution. We understand that some reservations led to a borderline reject rating, and hope to have dispelled any misgivings through this rebuttal, demonstrating the merit of our work and addressing any lingering doubts. If there are still concerns, we'll promptly respond to further queries.
>
> The authors

---

> > ### Comment · Reviewer_YEyS · 2023-08-14
> > **Response to rebuttal**
> >
> > I would like to thank the authors for clarifications.
> > I have carefully read the rebuttal and acknowledged that the proposed approach demonstrates strong performance.
> > I will reconsider the rating after discussion with other reviewers.

---

### Official Review · Reviewer_1MpK · 2023-07-05

**Soundness:** 3 good
**Presentation:** 3 good
**Contribution:** 3 good
**Rating:** 5
**Confidence:** 2

**Summary:**

This paper proposes two techniques to improve the performance of Monte-Carlo patch tracing on real-time image rendering: 1) keep all previously sampled values to improve spatial-temporal information reuse; 2) use reinforcement learning to optimize the sampling importance network, avoiding the explicit numerically approximated gradients.

**Strengths:**

The problem is clearly illustrated and the motivation is easy to understand. The experiments are adequate to demonstrate the arguments of this paper.

**Weaknesses:**

1. Line71 describes some methods that improve the spatiotemporal reuse by storing not only the averaged pxiel values. But there are no further discussion about their difference to the proposed method.

2. Although a well-trained RL framework performs good during inference, the training of a RL framework is usually unstable. The authors are suggested to add more details about the training techniques.



------------Post Rebutal-----------

I have carefully read other reviewers' comments and the authors' responses. Although I think this paper proposed a good method for the defined problem, other reviewers argue that most of the techniques used in this paper have been discussed in previous papers, which heavily weakens the novelty of this paper. Since I am not familiar with this topic, I adjust my score for the concern of the novelty.

**Questions:**

See the weakness.

**Limitations:**

There are no potential negative societal impacts of this work.

---

> ### Author Rebuttal · Authors · 2023-08-08
>
>
> Dear Reviewer 1MpK, thank you for your time and interesting remarks.
>
> > Line 71 describes some methods that improve the spatiotemporal reuse. But there are no further discussion about their difference to the proposed method.
>
> The mentioned methods are ReSTIR and derivative work of ReSTIR. Those methods do not use neural networks and are very different from ours:
>  Their latent space consists of storing a buffer of lights for every pixel. They update the buffer by sampling from new visible lights or reusing spatiotemporally nearby pixels, leveraging simple statistical heuristics (importance sampling). This latent space is directly used only for denoising.
> Differently, our latent space consists of storing a learned encoding given all previously available information to guide future sampling and denoising. It is updated using a dedicated neural network given new information and the previous state.
> We further discuss their conceptual and qualitative differences to our method in Figure 3 and lines [108-123, 266-276, 302-307].
>
>
>
> > Although a well-trained RL framework performs good during inference, the training of a RL framework is usually unstable. The authors are suggested to add more details about the training techniques.
>
> RL training is indeed often unstable. This is linked to several factors such as delayed rewards, non-stationarity, and off-policy learning. In our case, these factors are not as present as in other applications: rewards are delayed for very few frames/time steps as opposed to robotics applications; non-stationarity is present through the latent space, however, an effective latent space representation (i.e., our only system state) is not required to produce a first meaningful output and thus allows for a smooth instead of a steep learning curve; off-policy learning is commonly used as on-policy learning requires too many expensive simulation steps, however, as our environment is not as complex to simulate (in the order of tens of milliseconds; producing over 1M results/output values per step), we can apply on-policy learning. We further train only the sampling importance network using RL to overcome the hard/non-differentiable decision on how many samples to collect and use standard backpropagation for all other components. For those reasons, our training was relatively stable and converged with insignificant variance to the results presented. Here are other training parameters we tuned: buffer size of size 100, kl_coeff of weight 5e-7, kl_target of weight 5e-8, lambda of weight 0.2, vf_loss_coeff of 0.5, and an entropy_coeff of 1e-5.
>
> We hope that we provided satisfactory answers to your questions and comments, and that any remaining concerns could be addressed.
>
> The authors

---

### Official Review · Reviewer_WTeK · 2023-07-06

**Soundness:** 3 good
**Presentation:** 2 fair
**Contribution:** 2 fair
**Rating:** 5
**Confidence:** 1

**Summary:**

This paper tackles the issue of Monte-Carlo path tracing which is an important field for computer graphics and rendering. The paper first analyzes the current state of the art and identifies mainly two flaws which are addressed thereafter: First, the authors introduce a spatio temporal latent space serving as input to the denoising autoencoder which outputs the final image frames as well as (as a feedback) to the importance sampling network which outputs a distribution for rays per pixel. Second, as opposed to previous work, this distribution is not sampled from a sampling heatmap as in previous works but since this is problematic during training due to a coarse numerical estimation of the output gradient with respect to the gradient. Instead an RL based method for importance sampling is used during training which removes this need for a numerical estimation of the gradient. The resulting model seems to yield higher PSNR values than previous works.   However I am very unfamiliar with this topic so I cannot really estimate how thorough the evaluation is done.

**Strengths:**

* The paper seems to identify limitations and flaws in current methods which are then overcome by simple solutions which can furthermore be trained in combination in and end-to-end manner. However since I don't know all the related work, I could've missed sth.
* The proposed results seem to yield better results (measured by PSNR) at the cost of slightly higher inference times

**Weaknesses:**

* For Table 1, there is no information regarding the dataset and resolution of the images/videos used to assess the presented values. This is confusing, because the PSNR values are shown for 4 spp, resulting in an inference time of 22.5 ms which is faster than base. However, in Fig. 3, the average inference time for the proposed method  is much higher . Can the authors clarify this?
* The results are not presented as a function of image resolution (or number of pixels in the image). It would be interesting and important (e.g. to estimate the method's rea-world applicability) to report that, since nowadays most images/video to be rendered are of high spatial resolution.
* training datasets are nowhere listed? Is the method trained on all three presented datasets jointly or independently for each of them?
* Writing down the final objective function for the overall model would be good, since it would give the reader a final overview and summarize all the different aspects of the method.

**Questions:**

* Is PSNR the only important metric in this field? SSIM should be a comparable measure in my experience. Why is SSIM not reported?
* How does the method generalize to unseen data (not from validation/test sets). Did you try to apply it to some other data?

**Limitations:**

no limitations section is present. It would be imoportant to add such a section in my opinion.

---

> ### Author Rebuttal · Authors · 2023-08-08
>
> Dear Reviewer WTeK, thank you for your time and interesting remarks.
>
> > For Table 1, there is no information regarding the dataset and resolution of the images/videos used to assess the presented values. This is confusing, because the PSNR values are shown for 4 spp, resulting in an inference time of 22.5 ms. However, in Fig. 3, the average inference time for the proposed method is much higher.
>
> In Table 1, we report the aggregate inference time of the neural networks or the method in general for related work. We exclude the pixel sampling time (thus also the 0.0 inference time for Monte-Carlo sampling). Differently, in Table 3, we report the entire frame latency of the whole frameworks including the pixel sampling, showing that our method is effectively faster. In terms of evaluation data, we use the procedure outlined in Section 3.1: Experimental Setup, i.e., the 4 mentioned scenes with leave-one-scene-out cross-validation.
>
> > The results are not presented as a function of image resolution
>
> We trained and evaluated using resolutions of 720x720 pixels (cf. Line 233), as in OIDN, and more than in other related work (DASR: 512x512). Our method scales linearly in terms of inference time with the number of pixels for higher resolutions. For high-resolution rendering, the outputs of ray-tracing methods such as ours as well as related work are commonly applied at a below-native resolution and upsampled/interpolated using methods such as Nvidia's DLSS or AMD's FidelityFX Super Resolution as a post-processing step. An accurate estimate for your preferred resolution can thus be obtained by scaling the reported time proportionally to the ratio between the desired pixel count (after considering the preferred upscaling factor) and $720^2$. The relative latency to related work will remain unchanged.
>
> > Training datasets are nowhere listed? Is the method trained on all three presented datasets jointly or independently for each of them?
>
> There are no commonly used datasets available for evaluating ray tracing methods. However, several scenes are available with licenses compatible to perform such research. We list the 4 scenes used in this work in Section 3.1 Experimental Setup (cf lines 211-213), and further release the code to reproduce the results in the supplementary materials. For the quantitative evaluations, we perform 4-fold cross-validation across the scenes (cf. lines 235-236), i.e., we pick one scene for testing and train on the remaining 3 scenes, then rotate to the next scenes for evaluation while averaging the results.
>
> > Writing down the final objective function for the overall model would be good, since it would give the reader a final overview and summarize all the different aspects of the method.
>
> If we understand the remark correctly, this is presented in Line 192.
>
> > Is PSNR the only important metric in this field? SSIM should be a comparable measure in my experience. Why is SSIM not reported?
>
> We agree that SSIM is the better metric as it is closer to visual perception, and thus we use its multi-scale variant in our loss function (Line 186). We report PSNR as the metric for comparability as all related works (DASR, NTAS, ReSTIR, OIDN, ...) evaluate with PSNR.
>
> > How does the method generalize to unseen data (not from validation/test sets). Did you try to apply it to some other data?
>
> The test data is unseen. While we would like to expand both the training and test sets, the available data with a license permitting research use is very limited and most related work trains and evaluates on non-public data, making the results non-reproducible. We aim to improve on this by releasing code of not just our own method but also re-implementations of related work, with consistent evaluations on the same data.
>
> > No limitations section is present. It would be important to add such a section in my opinion.
>
> If permitted, we will add the following section to the paper or at least to the appendix.
>
> ```
> Limitations: The current method does not consider the application of after-effects such as motion blur, which could allow the reduction of samples collected on fast-moving objects, while in the current setting we would implicitly focus the ray tracing samples on such an object to minimize the error. Further, this method is applicable to entertainment applications and can potentially generate artifacts not present in the real scene, which could be an obstacle in application scenarios such as VR/AR medical devices. Additionally, it requires a system capable of performing both path tracing and DNN inference. While current graphics cards provide this capability and we assess the frame rendering time considering both components, the underlying workloads are significantly different: DNN inference is generally a very structured and high arithmetic intensity workload whereas path tracing is branching-intensive and performs random look-ups into memory.  In future systems, it is conceivable that dedicated devices are used for each step, which would enforce a fixed capability for each type of compute and limit a free trade-off between the two as we make use in this work. Additionally, applying a DNN adds a memory overhead, although it can remain minimal compared to other components such as textures that commonly fill most the GPU's memory. Specifically, the model requires 110 MB to store the input and latent space data, <50 MB to store the model, and a few MB of working memory for intermediate feature maps that can be processed in tiles.  This overhead is insignificant (<2%) compared to >12GB of textures loaded for current games on high-end GPUs where such capacity is available.
> ```
>
> We appreciate your dedication to thoroughly review NeurIPS papers including ours. We hope to have cleared out all or most of your concerns and would be delighted to see a revision of the rating. If any questions or doubts remain, we will gladly answer any further questions.
>
> The authors

---

### Official Review · Reviewer_CLYz · 2023-07-07

**Soundness:** 3 good
**Presentation:** 3 good
**Contribution:** 3 good
**Rating:** 6
**Confidence:** 3

**Summary:**

This paper proposes to use reinforcement learning (RL) to improve adaptive sampling effectiveness in Monte Carlo ray tracing.  Another key contribution they claim is the use of a latent space representation to encode temporal information, which improves the reuse of spatiotemporal pixel information across frames.  The impact of this work is to improve quality of real-time path tracing results with respect to noise and temporal stability.

**Strengths:**

The paper is well written, and provides a clear preliminary of the necessary background on adaptive sampling and denoising, as well as reviewing current SOTA approaches which are compared with their method.  The usage of the "spatiotemporal" latent
space, which is claimed to encode temporal information more effectively than previous works, is an insightful idea.  The RL-based, learned importance sampler is also an interesting approach that addresses limitations of prior works.  Qualitatively and quantitatively, their method outperforms other methods by a significant margin.

**Weaknesses:**

Evaluation of the method is limited to only a small number of scenes.  It would be nice to see a larger scale evaluation, though it is understandable that training/evaluation data is laborious to assemble.

Possible lack of theoretical novelty -- the method is validated with empirical results which are quite promising, but there is little discussion or analysis of the stability / robustness of the method.

Additional images for qualitative comparison could be helpful for visualizing some of the key concepts discussed in the paper.  For example, visualizing qualitative differences between various sample counts, or dedicated visualizations of the estimated sample maps and/or RL state.

**Questions:**

In Figure 2 and through the paper, some of the ground truth rendered images appear very dark.  Is this intentional and if so, why?

How sensitive is the inference performance to the choice of training data?  Are there any challenges generalizing to new rendered scenes?

This paper reviews the approach for approximating gradients from "Deep Adaptive Sampling for Low Sample Count Rendering (Kuznetsov et al.)", and proposes RL as an alternative to their formulation.  Out of curiosity, would integrating a differentiable renderer (e.g. Mitsuba) into the pipeline from Kuznetsov et al. provide any useful additional gradients?

Could you elaborate on why RL specifically is a good learning paradigm for learning the importance sampling module?  Does using RL

**Limitations:**

The paper does not identify any limitations, and I do not see any clear limitations.

---

> ### Author Rebuttal · Authors · 2023-08-08
>
> Dear Reviewer CLYz, thank you for your time and interesting remarks.
>
> > Additional images for qualitative comparison could be helpful for visualizing some of the key concepts discussed in the paper. For example, visualizing qualitative differences between various sample counts, or dedicated visualizations of the estimated sample maps and/or RL state.
>
> We provide qualitative comparisons and estimated sample maps for various sample counts in the supplementary material as well as some additional visualizations: state (stt.png), sample maps (hitmap.png, 540hitmap.png) and comparisons for various sample counts (0.1spp.gif, 1.0spp.gif, 4.0spp.gif) in the following anonymous link: https://drive.google.com/drive/folders/1zrKGelIwSxcGkww6u3-MYDRyVTO45SJ3?usp=sharing . We will include them all in the final supplementary materials.
>
> > How sensitive is the inference performance to the choice of training data? Are there any challenges generalizing to new rendered scenes?
>
> Our main results are obtained by performing a Leave One Scene Out Validation (See line 235). This means that given one test scene, we train on all scenes except the test scene, and then evaluate on the test scene. We repeat the same procedure such that every scene becomes the test scene once and finally average results. Our results show low sensitivity to the choice of training data (see appendix for individual Leave One Scene Out results) and good generalization to the unseen data from the test scene.
>
> > In Figure 2 and through the paper, some of the ground truth rendered images appear very dark. Is this intentional and if so, why?
>
> The ground truth images selected for visualization in the paper are not outliers in terms of brightness. We agree that two of the four images appear dark in the paper. However, zooming in on them in full-screen without any of the paper's white background visible as to mimick an in-game experience will change the perception to this being a rather "normal" brightness.
> We do not feel comfortable performing range compression or otherwise modifying the data for an improved visual appearance of the ground truth frames in the figures in the paper.
>
> > This paper reviews the approach for approximating gradients from "Deep Adaptive Sampling for Low Sample Count Rendering (Kuznetsov et al.)", and proposes RL as an alternative to their formulation. Out of curiosity, would integrating a differentiable renderer (e.g. Mitsuba) into the pipeline from Kuznetsov et al. provide any useful additional gradients?
>
> While the renderer would become differentiable, the hard/non-differentiable decision on how many samples to collect would remain. It would open up the possibility to learn a continuous-valued sampling recommendation which could be quantized to integer values with a straight-through estimator, stochastic rounding, or similar way during the forward pass and inaccurate gradients in the backward pass. However, it is not immediately obvious how this would impact the results, i.e., if there would be a benefit to dropping the RL component in favor of inaccurate gradients (straight-through estimator, stochastic rounding). In any case, it would be impractical, as artists develop scenes for a specific game engine that is then the only program that can render the desired output, which generally does not support differentiable rendering.
>
> > Could you elaborate on why RL specifically is a good learning paradigm for learning the importance sampling module?
>
> Quantized problems, such as choosing an integer number of samples to collect per pixel, are commonly addressed by using hard decisions or stochastic rounding in the forward step and computing (very) approximate gradients for the backward pass (e.g., straight-through estimator). This negatively impacts the learning process and quality of results. We can avoid such an approximation and instead formulate an exact end-to-end optimization criterion using reinforcement learning. Using RL further allows collecting less than 1spp, which is a common limitation in related work that strongly impacts the capabilities in low-latency/real-time rendering scenarios. Permitting to collect 0 samples at some pixels with gradient-based learning implies that there is no longer a gradient to the sampling recommendation network and it can thus not be learned appropriately for this scenario.
>
> We hope that we provided satisfactory answers to your questions and comments, and that any remaining concerns could be addressed.
>
> The authors

---

> > ### Comment · Reviewer_CLYz · 2023-08-16
> >
> > Thanks to the authors for answering my questions.  I have read the rebuttal and acknowledge your clarifications.
> >
> > I believe one of this paper's main contributions is the design of a system that produces results that are superior to other methods empirically.  As such, my main remaining concern is the lack of more extensive experimental validation of the results.  Some of the closely-related works that are mentioned evaluate their methods on a greater diversity of scenes.  I have viewed all of the supplementary content, and believe that more extensive evaluation would make this work more convincing.
> >
> > I will keep my rating the same for now.

---

### Author Rebuttal · Authors · 2023-08-08

Based on the feedback of Reviewers WTeK and YEyS, and if permitted, we will add the following section to the paper or at least to the appendix. Please look at individual rebuttals for all other comments and answers.

```
Limitations: The current method does not consider the application of after-effects such as motion blur, which could allow the reduction of samples collected on fast-moving objects, while in the current setting we would implicitly focus the ray tracing samples on such an object to minimize the error. Further, this method is applicable to entertainment applications and can potentially generate artifacts not present in the real scene, which could be an obstacle in application scenarios such as VR/AR medical devices. Additionally, it requires a system capable of performing both path tracing and DNN inference. While current graphics cards provide this capability and we assess the frame rendering time considering both components, the underlying workloads are significantly different: DNN inference is generally a very structured and high arithmetic intensity workload whereas path tracing is branching-intensive and performs random look-ups into memory. As these are fundamentally different, we see dedicated ray tracing units on modern GPUs. In future systems, it is conceivable that dedicated devices are used for each step, which would enforce a fixed capability for each type of compute and limit a free trade-off between the two as we make use in this work. Additionally, applying a DNN adds a memory overhead, although it can remain minimal compared to other components such as textures that commonly fill most the GPU's memory. Specifically, the model requires 110 MB more memory to store the input and latent space data, <50 MB to store the model, and a few MB of working memory for intermediate feature maps that can be processed in tiles.

Memory overhead: We store the following data in memory. 1) the model weights (<50MB), 2) the sampled pixel values (24 channels; RGB values for up to 8 non-averaged samples), 3) 7 additional input channels (3 for surface normals, 3 for albedo, 1 for depth), 4) 32 channels for the state (warped latent space), and 5) a small working memory for intermediate feature data during inference. The components 2-4 add up to 53 more channels than previous works (which use 3 channels for input channels and 7 for additional data); hence 212 Byte/pixel; 110MB in total for 720x720 pixel data; and the required working memory is minimal as the inference can be done on tiles. With a total memory footprint in the order of 200 MB, the overhead is insignificant (<2%) compared to >12GB of textures loaded for current games on high-end GPUs where such capacity is available.
```
We hope that any remaining concerns could be addressed.

The authors

---

### Decision · Program_Chairs · 2023-09-21

**Decision:**

Accept (poster)

**Comment:**

After considering the rebuttal and internal discussions, all reviewers are in favor of accepting this submission. The paper is well-written, it identifies open issues in previous work, and presents simple solutions that are empirically shown to be effective (though experiments could be extended)
-> Accept. Congratulations!
Please take the reviewer comments into account when working on the camera-ready.